# Adaptive Extra-Gradient Methods for Min-Max Optimization and Games

**Kimon Antonakopoulos**
Univ. Grenoble Alpes, CNRS, Inria, Grenoble INP
LIG, 38000 Grenoble, France
kimon.antonakopoulos@inria.fr

**E. Veronica Belmega**
ETIS/ENSEA
Univ. de Cergy-Pontoise-CNRS, France
belmega@ensea.fr

**Panayotis Mertikopoulos**
Univ. Grenoble Alpes, CNRS, Inria, Grenoble INP, LIG, 38000 Grenoble, France &
Criteo AI Lab
panayotis.mertikopoulos@imag.fr

## Abstract

We present a new family of min-max optimization algorithms that automatically exploit the geometry of the gradient data observed at earlier iterations to perform more informative extra-gradient steps in later ones. Thanks to this adaptation mechanism, the proposed method automatically detects whether the problem is smooth or not, without requiring any prior tuning by the optimizer. As a result, the algorithm simultaneously achieves order-optimal convergence rates, i.e., it converges to an $\varepsilon$-optimal solution within $\mathcal{O}(1/\varepsilon)$ iterations in smooth problems, and within $\mathcal{O}(1/\varepsilon^2)$ iterations in non-smooth ones. Importantly, these guarantees do not require any of the standard boundedness or Lipschitz continuity conditions that are typically assumed in the literature; in particular, they apply even to problems with singularities (such as resource allocation problems and the like). This adaptation is achieved through the use of a geometric apparatus based on Finsler metrics and a suitably chosen mirror-prox template that allows us to derive sharp convergence rates for the methods at hand.

## 1 Introduction

The surge of recent breakthroughs in generative adversarial networks (GANs) [20], robust reinforcement learning [41], and other adversarial learning models [27] has sparked renewed interest in the theory of min-max optimization problems and games. In this broad setting, it has become empirically clear that, ceteris paribus, the simultaneous training of two (or more) antagonistic models faces drastically new challenges relative to the training of a single one. Perhaps the most prominent of these challenges is the appearance of cycles and recurrent (or even chaotic) behavior in min-max games. This has been studied extensively in the context of learning in bilinear games, in both continuous [16, 31, 40] and discrete time [12, 18, 19, 32], and the methods proposed to overcome recurrence typically focus on mitigating the rotational component of min-max games. The method with the richest history in this context is the *extra-gradient* (EG) algorithm of Korpelevich [25] and its variants. The EG algorithm exploits the Lipschitz smoothness of the problem and, if coupled with a Polyak–Ruppert averaging scheme, it achieves an $\mathcal{O}(1/T)$ rate of convergence in smooth, convex-concave min-max problems [35]. This rate is known to be tight [34, 39] but, in order to achieve it, the original method requires the problem's Lipschitz constant to be known in advance. If the problem is not Lipschitz smooth (or the algorithm is run with a vanishing step-size schedule), the method's rate of convergence drops to $\mathcal{O}(1/\sqrt{T})$.

**Our contributions.** Our aim in this paper is to provide an algorithm that automatically adapts to smooth / non-smooth min-max problems and games, and achieves order-optimal rates in both classes without requiring any prior tuning by the optimizer. In this regard, we propose a flexible algorithmic scheme, which we call AdaProx, and which exploits gradient data observed at earlier iterations to perform more informative extra-gradient steps in later ones. Thanks to this mechanism, and to the best of our knowledge, AdaProx is the first algorithm that simultaneously achieves the following:

|  | EG [24, 25, 35] | GRAAL [29] | GMP [47] | AMP [1, 17] | BL [2] | ADAPROX [OURS] |
|---|---|---|---|---|---|---|
| PARAMETER-AGNOSTIC | ✗ | ✓ | PARTIAL | ✓ | PARTIAL | ✓ |
| RATE INTERPOLATION | ✗ | ✗ | ✓ | ✗ | ✓ | ✓ |
| UNBOUNDED DOMAIN | ✗ | ✓ | ✗ | ✗ | ✗ | ✓ |
| SINGULARITIES | ✗ | ✗ | ✗ | ✓ | ✗ | ✓ |

**Table 1:** Overview of related work. For the purposes of this table, "parameter-agnostic" means that the method does not require prior knowledge of the parameters of the problem it was designed to solve (Lipschitz modulus, domain diameter, etc.); "rate interpolation" means that the algorithm's convergence rate is $\mathcal{O}(1/T)$ or $\mathcal{O}(1/\sqrt{T})$ in smooth/non-smooth problems respectively; "unbounded domain" is self-explanatory; and, finally, "singularities" means that the problem's defining vector field may blow up at a boundary point of the problem's domain.

1. An $\mathcal{O}(1/\sqrt{T})$ convergence rate in non-smooth problems and $\mathcal{O}(1/T)$ in smooth ones.

2. Applicability to min-max problems and games where the standard boundedness / Lipschitz continuity conditions required in the literature do not hold.

3. Convergence without prior knowledge of the problem's parameters (e.g., whether the problem's defining vector field is smooth or not, its smoothness modulus if it is, etc.).

Our proposed method achieves the above by fusing the following ingredients: *a*) a family of local norms – a *Finsler metric* – capturing any singularities in the problem at hand; *b*) a suitable mirror-prox template; and *c*) an adaptive step-size policy in the spirit of Rakhlin & Sridharan [43]. We also show that, under a suitable coherence assumption, the sequence of iterates generated by the algorithm converges, thus providing an appealing alternative to iterate averaging in cases where the method's "last iterate" is more appropriate (for instance, if using AdaProx to solve non-monotone problems).

**Related works.** There have been several works improving on the guarantees of the original extra-gradient/mirror-prox template. We review the most relevant of these works below; for convenience, we also tabulate these contributions in Table 1 above. Because many of these works appear in the literature on variational inequalities [15], we also use this language in the sequel. In unconstrained problems with an operator that is locally Lipschitz continuous (but not necessarily globally so), the *golden ratio algorithm* (GRAAL) [29] achieves convergence without requiring prior knowledge of the problem's Lipschitz parameter. However, GRAAL provides no rate guarantees for non-smooth problems – and hence, a fortiori, no interpolation guarantees either. By contrast, such guarantees are provided in problems with a bounded domain by the *generalized mirror-prox* (GMP) algorithm of [47] under the umbrella of Hölder continuity. Still, nothing is known about the convergence of GRAAL/GMP in problems with singularities (i.e., when the problem's defining vector field blows up at a boundary point of the problem's domain). Singularities of this type were treated in a recent series of papers [1, 17, 48] by means of a "Bregman continuity" or "Lipschitz-like" condition. These methods are order-optimal in the smooth case, without requiring any knowledge of the problem's smoothness modulus. On the other hand, like GRAAL (but unlike GMP), they do not provide any rate interpolation guarantees between smooth and non-smooth problems. Another method that simultaneously achieves an $\mathcal{O}(1/\sqrt{T})$ rate in non-smooth problems and an $\mathcal{O}(1/T)$ rate in smooth ones is the recent algorithm of Bach & Levy [2]. The BL algorithm employs an adaptive, AdaGrad-like step-size policy which allows the method to interpolate between the two regimes – and this, even with noisy gradient feedback. On the negative side, the BL algorithm requires a bounded domain with a (Bregman) diameter that is known in advance; as a result, its theoretical guarantees do not apply to unbounded problems. In addition, the BL algorithm makes crucial use of boundedness and Lipschitz continuity; extending the BL method beyond this standard framework is a highly non-trivial endeavor which formed a big part of this paper's motivation.

## 2  PROBLEM SETUP AND BLANKET ASSUMPTIONS

We begin in this section by reviewing some basics for min-max problems and games.

**2.1. Min-max / Saddle-point problems.** A *min-max game* is a saddle-point problem of the form

$$\min_{\theta \in \Theta} \max_{\phi \in \Phi} \mathcal{L}(\theta, \phi) \tag{SP}$$

where $\Theta$, $\Phi$ are convex subsets of some ambient real space and $\mathcal{L}\colon \Theta \times \Phi \to \mathbb{R}$ is the problem's *loss function*. In the game-theoretic interpretation of (SP), the player controlling $\theta$ seeks to minimize $\mathcal{L}(\theta, \phi)$ for any value of the maximization variable $\phi$, while the player controlling $\phi$ seeks to maximize $\mathcal{L}(\theta, \phi)$ for any value of the minimization variable $\theta$. Accordingly, solving (SP) consists of finding a *Nash equilibrium* (NE), i.e., an action profile $(\theta^*, \phi^*) \in \Theta \times \Phi$ such that

$$\mathcal{L}(\theta^*, \phi) \leq \mathcal{L}(\theta^*, \phi^*) \leq \mathcal{L}(\theta, \phi^*) \quad \text{for all } \theta \in \Theta, \phi \in \Phi. \tag{1}$$

By the minimax theorem of von Neumann [49], Nash equilibria are guaranteed to exist when $\Theta$, $\Phi$ are compact and $\mathcal{L}$ is convex-concave (i.e., convex in $\theta$ and concave in $\phi$). Much of our paper is motivated by the question of calculating a Nash equilibrium $(\theta^*, \phi^*)$ of (SP) in the context of von Neumann's theorem; we expand on this below.

**2.2. Games.** Going beyond the min-max setting, a *continuous game in normal form* is defined as follows: First, consider a finite set of players $\mathcal{N} = \{1, \ldots, N\}$, each with their own action space $\mathcal{K}_i \in \mathbb{R}^{d_i}$ (assumed convex but possibly not closed). During play, each player selects an action $x_i$ from $\mathcal{K}_i$ with the aim of minimizing a loss determined by the ensemble $x \coloneqq (x_i; x_{-i}) \coloneqq (x_1, \ldots, x_N)$ of all players' actions. In more detail, writing $\mathcal{K} \coloneqq \prod_i \mathcal{K}_i$ for the game's total action space, we assume that the loss incurred by the $i$-th player is $\ell_i(x_i; x_{-i})$, where $\ell_i\colon \mathcal{K} \to \mathbb{R}$ is the player's *loss function*. In this context, a Nash equilibrium is any action profile $x^* \in \mathcal{K}$ that is *unilaterally stable*, i.e.,

$$\ell_i(x_i^*; x_{-i}^*) \leq \ell_i(x_i; x_{-i}^*) \quad \text{for all } x_i \in \mathcal{K}_i \text{ and all } i \in \mathcal{N}. \tag{NE}$$

If each $\mathcal{K}_i$ is compact and $\ell_i$ is convex in $x_i$, existence of Nash equilibria is guaranteed by the theorem of Debreu [13]. Given that a min-max problem can be seen as a two-player zero-sum game with $\ell_1 = \mathcal{L}$, $\ell_2 = -\mathcal{L}$, von Neumann's theorem may in turn be seen as a special case of Debreu's; in the sequel, we describe a first-order characterization of Nash equilibria that encapsulates both. In most cases of interest, the players' loss functions are *individually subdifferentiable* on a subset $\mathcal{X}$ of $\mathcal{K}$ with $\mathrm{ri}\,\mathcal{K} \subseteq \mathcal{X} \subseteq \mathcal{K}$ [21, 44]. This means that there exists a (possibly discontinuous) vector field $V_i\colon \mathcal{X} \to \mathbb{R}^{d_i}$ such that

$$\ell_i(x_i'; x_{-i}) \geq \ell_i(x_i; x_{-i}) + \langle V_i(x), x_i' - x_i \rangle \tag{2}$$

for all $x \in \mathcal{X}$, $x' \in \mathcal{K}$ and all $i \in \mathcal{N}$ [21]. In the simplest case, if $\ell_i$ is differentiable at $x$, then $V_i(x)$ can be interpreted as the gradient of $\ell_i$ with respect to $x_i$. The *raison d'être* of the more general definition (2) is that it allows us to treat non-smooth loss functions that are common in machine learning (such as $L^1$-regularized losses). We make this distinction precise below:

1. If there is no continuous vector field $V_i(x)$ satisfying (2), the game is called *non-smooth*.

2. If there is a continuous vector field $V_i(x)$ satisfying (2), the game is called *smooth*.

*Remark.* We stress here that the adjective "smooth" refers to the game itself: for instance, if $\ell(x) = |x|$ for $x \in \mathbb{R}$, the game is not smooth and any $V$ satisfying (2) is discontinuous at 0. In this regard, the above boils down to whether the (individual) subdifferential of each $\ell_i$ admits a continuous selection.

**2.3. Resource allocation and equilibrium problems.** The notion of a Nash equilibrium captures the unilateral minimization of the players' individual loss functions. In many pratical cases of interest, a notion of equilibrium is still relevant, even though it is not necessarily attached to the minimization of individual loss functions. Such problems are known as "equilibrium problems" [15, 26]; to avoid unnecessary generalities, we focus here on a relevant problem that arises in distributed computing architectures (such as GPU clusters and the like). To state the problem, consider a distributed computing grid consisting of $N$ parallel processors that serve demands arriving at a rate of $\rho$ per unit of time (measured e.g., in flop/s). If the maximum processing rate of the $i$-th node is $\mu_i$ (without overclocking), and jobs are buffered and served on a first-come, first-served (FCFS) basis, the mean time required to process a unit demand at the $i$-th node is given by the Kleinrock M/M/1 response function $\tau_i(x_i) = 1/(\mu_i - x_i)$, where $x_i$ denotes the node's *load* [5]. Accordingly, the set of feasible loads that can be processed by the grid is $\mathcal{X} \coloneqq \{(x_1, \ldots, x_N) : 0 \leq x_i < \mu_i, x_1 + \cdots + x_N = \rho\}$. In this context, a load profile $x^* \in \mathcal{X}$ is said to be *balanced* if no infinitesimal process can be better served by buffering it at a different node [38]; formally, this amounts to the so-called *Wardrop equilibrium* condition

$$\tau_i(x_i^*) \leq \tau_j(x_j^*) \quad \text{for all } i, j \in \mathcal{N} \text{ with } x_i^* > 0. \tag{WE}$$

We note here a crucial difference between (WE) and (NE): if we view the grid's computing nodes as "players", the constraint $\sum_i x_i = \rho$ means that there is no allowable unilateral deviation $(x_i^*; x_{-i}^*) \mapsto (x_i; x_{-i}^*)$ with $x_i \neq x_i^*$. As a result, (NE) is meaningless as a requirement for this equilibrium problem.

As we discuss below, this resource allocation problem will require the full capacity of our framework.

**2.4. Variational inequalities.** Importantly, all of the above problems can be restated as a *variational inequality* of the form

$$\text{Find } x^* \in \mathcal{X} \text{ such that } \langle V(x^*), x - x^* \rangle \geq 0 \text{ for all } x \in \mathcal{X}. \tag{VI}$$

In the above, $\mathcal{X}$ is a convex subset of $\mathbb{R}^d$ (not necessarily closed) that represents the problem's *domain*. The problem's *defining vector field* $V \colon \mathcal{X} \to \mathbb{R}^d$ is then given as follows: In min-max problems and games, $V$ is any field satisfying (2); otherwise, in equilibrium problems of the form (WE), the components of $V$ are $V_i = \tau_i$ (we leave the details of this verification to the reader). This equivalent formulation is quite common in the literature on min-max / equilibrium problems [14, 15, 26, 30], and it is often referred to as the "vector field formulation" [3, 8, 23]. Its usefulness lies in that it allows us to abstract away from the underlying game-theoretic complications (multiple indices, individual subdifferentials, etc.) and provides a unifying framework for a wide range of problems in machine learning, signal processing, operations research, and many other fields [15, 45]. For this reason, our analysis will focus almost exclusively on solving (VI), and we will treat $V$ and $\mathcal{X} \subseteq \mathbb{R}^d$, $d = \sum_i d_i$, as the problem's primitive data.

**2.5. Merit functions and monotonicity.** A widely used assumption in the literature on equilibrium problems and variational inequalities is the *monotonicity condition*

$$\langle V(x) - V(x'), x - x' \rangle \geq 0 \quad \text{for all } x, x' \in \mathcal{X}. \tag{Mon}$$

In single-player games, monotonicity is equivalent to convexity of the optimizer's loss function; in min-max games, it is equivalent to $\mathcal{L}$ being convex-concave [26]; etc. In the absence of monotonicity, approximating an equilibrium is PPAD-hard [11], so we will state most of our results under (Mon).

Now, to assess the quality of a candidate solution $\hat{x} \in \mathcal{X}$, we will employ the *restricted merit function*

$$\text{Gap}_{\mathcal{C}}(\hat{x}) = \sup_{x \in \mathcal{C}} \langle V(x), \hat{x} - x \rangle, \tag{3}$$

where the "*test domain*" $\mathcal{C}$ is a nonempty convex subset of $\mathcal{X}$ [15, 24, 37]. The motivation for this is provided by the following proposition:

**Proposition 1.** *Let $\mathcal{C}$ be a nonempty convex subset of $\mathcal{X}$. Then: a) $\text{Gap}_{\mathcal{C}}(\hat{x}) \geq 0$ whenever $\hat{x} \in \mathcal{C}$; and b) if $\text{Gap}_{\mathcal{C}}(\hat{x}) = 0$ and $\mathcal{C}$ contains a neighborhood of $\hat{x}$, then $\hat{x}$ is a solution of* (VI).

Proposition 1 generalizes an earlier characterization by Nesterov [37] and justifies the use of $\text{Gap}_{\mathcal{C}}(x)$ as a merit function for (VI); to streamline our presentation, we defer the proof to the paper's supplement. Moreover, to avoid trivialities, we will also assume that the solution set $\mathcal{X}^*$ of (VI) is nonempty and we will reserve the notation $x^*$ for solutions of (VI). Together with monotonicity, this will be our only blanket assumption.

## 3 The Extra-Gradient Algorithm and its Limits

Perhaps the most widely used solution method for games and variational inequalities (VIs) is the *extra-gradient* (EG) algorithm of Korpelevich [25] and its variants [28, 42, 43]. This algorithm has a rich history in optimization, and it has recently attracted considerable interest in the fields of machine learning and AI, see e.g., [8, 12, 18, 22, 23, 32, 33] and references therein.

In its simplest form, for problems with closed domains, the algorithm proceeds recursively as

$$X_{t+1/2} = \Pi(X_t - \gamma_t V_t), \qquad X_{t+1} = \Pi(X_t - \gamma_t V_{t+1/2}), \tag{EG}$$

where $\Pi(x) = \arg\min_{x' \in \mathcal{X}} \|x' - x\|$ is the Euclidean projection on $\mathcal{X}$, $V_t \coloneqq V(X_t)$ for $t = 1, 3/2, \ldots$, and $\gamma_t > 0$, is the method's step-size. Then, running (EG) for $T$ iterations, the algorithm returns the "ergodic average"

$$\bar{X}_T = \frac{\sum_{t=1}^{T} \gamma_t X_{t+1/2}}{\sum_{t=1}^{T} \gamma_t}. \tag{4}$$

In this setting, the main guarantees for (EG) date back to [35] and can be summarized as follows:

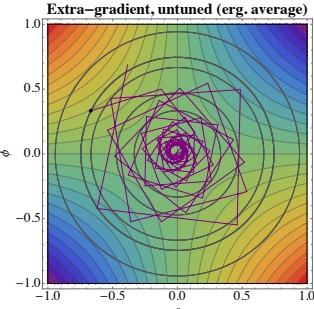 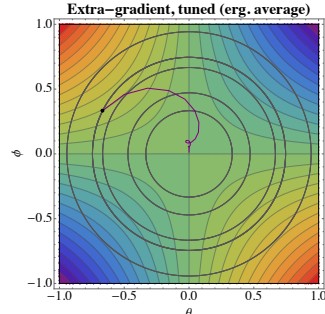 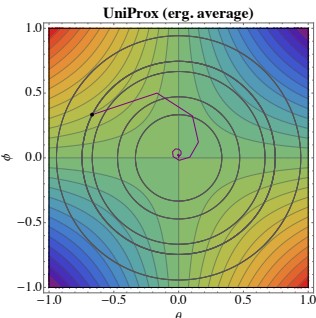

**Figure 1:** The behavior of (EG) in the bilinear min-max problem $\mathcal{L}(\theta, \phi) = \theta\phi$ with $\theta, \phi \in [-1, 1]$. Given the clipping at $[-1, 1]$, this problem is smooth with $L = 1$; instead, in the unconstrained case, both (BD) and (LC) fail. Still, even in the constrained case, running (EG) with a step-size only slightly above the $1/L$ bound ($L = 1$, $\gamma = 1.04$) results in a dramatic convergence failure (left plot). Tuning the step-size of (EG) resolves this problem (center), but a constant step-size makes the algorithm unnecessarily conservative towards the end. The proposed AdaProx algorithm automatically exploits previous gradient data to perform more informative extra-gradient steps in later ones, thus achieving faster convergence without tuning.

1. *For non-smooth problems* (discontinuous $V$): Assume $V$ is *bounded*, i.e., there exists some $M > 0$ such that

$$\|V(x)\| \leq M \quad \text{for all } x \in \mathcal{X}. \tag{BD}$$

Then, if (EG) is run with a step-size of the form $\gamma_t \propto 1/\sqrt{t}$, we have

$$\text{Gap}_{\mathcal{C}}(\bar{X}_T) = \mathcal{O}(1/\sqrt{T}). \tag{5}$$

2. *For smooth problems* (continuous $V$): Assume $V$ is *L-Lipschitz continuous*, i.e.,

$$\|V(x) - V(x')\| \leq L\|x - x'\| \quad \text{for all } x, x' \in \mathcal{X}. \tag{LC}$$

Then, if (EG) is run with a constant step-size $\gamma < 1/L$, we have

$$\text{Gap}_{\mathcal{C}}(\bar{X}_T) = \mathcal{O}(1/T). \tag{6}$$

*Remark.* In the above, $\|\cdot\|$ is tacitly assumed to be the standard Euclidean norm. Non-Euclidean considerations will play a crucial role in the sequel, but they are not necessary for the moment.

Importantly, the distinction between smooth and non-smooth problems cannot be lifted: the bounds (5) and (6) are tight in their respective problem classes and they cannot be improved without further assumptions [34, 39]. Moreover, we should also note the following:

1. The algorithm changes drastically from the non-smooth to the smooth case: non-smoothness requires $\gamma_t \propto 1/\sqrt{t}$, but such a step-size cannot achieve a fast $\mathcal{O}(1/T)$ rate.

2. If (EG) is run with a constant step-size, $L$ must be known in advance; otherwise, running (EG) with an ill-adapted step-size ($\gamma > 1/L$) could lead to non-convergence.

We illustrate this failure of (EG) in Fig. 1. As we discussed in the introduction, our aim in the sequel will be to provide a single, *adaptive* algorithm that simultaneously achieves the following: *a*) an order-optimal $\mathcal{O}(1/\sqrt{T})$ convergence rate in non-smooth problems and $\mathcal{O}(1/T)$ in smooth ones; *b*) convergence in problems where the boundedness / Lipschitz continuity conditions (BD) / (LC) no longer hold; and *c*) achieves all this without prior knowledge of the problem's parameters.

## 4 RATE INTERPOLATION: THE EUCLIDEAN CASE

As a prelude to our main result, we provide in this section an adaptive version of (EG) that achieves the "best of both worlds" in the Euclidean setting of Section 3, i.e., an $\mathcal{O}(1/\sqrt{T})$ convergence rate in problems satisfying (BD), and an $\mathcal{O}(1/T)$ rate in problems satisfying (LC). Our starting point is the observation that, if the sequence $X_t$ produced by (EG) converges to a solution of (VI), the difference

$$\delta_t := \|V_{t+1/2} - V_t\| = \|V(X_{t+1/2}) - V(X_t)\| \tag{7}$$

must itself become vanishingly small if $V$ is (Lipschitz) continuous. On the contrary, if $V$ is *discontinuous*, this difference may remain bounded away from zero (consider for example the $L^1$ loss $\ell(x) = |x|$ near 0). Based on this observation, we consider the adaptive step-size policy:

$$\gamma_{t+1} = 1 \Big/ \sqrt{1 + \sum_{s=1}^{t} \delta_s^2}. \tag{8}$$

The intuition behind (8) is as follows: If $V$ is not smooth and $\liminf_{t\to\infty} \delta_t > 0$, then $\gamma_t$ will vanish at a $\Theta(1/\sqrt{t})$ rate, which is the optimal step-size schedule for problems satisfying (BD) but not (LC). Instead, if $V$ satisfies (LC) and $X_t$ converges to a solution $x^*$ of (VI), it is plausible to expect that the infinite series $\sum_t \delta_t^2$ is summable, in which case the step-size $\gamma_t$ will not vanish as $t \to \infty$. Furthermore, since $\delta_t$ is defined in terms of successive gradient differences, it automatically exploits the variation of the gradient data observed up to time $t$, so it can be expected to adjust to the "local" Lipschitz constant of $V$ around a solution $x^*$ of (VI).

Our step-size policy and motivation are similar in spirit to the "predictable sequence" approach of [43]. For now, we only state (without proof) our main result for problems satisfying (BD) or (LC).

**Theorem 1.** *Suppose $V$ satisfies* (Mon)*, let $\mathcal{C}$ be a compact neighborhood of a solution of* (VI)*, and let $H = \sup_{x\in\mathcal{C}} \|X_1 - x\|^2$. If* (EG) *is run with the adaptive step-size policy* (8)*, we have:*

*a) If $V$ satisfies* (BD)*:*   $\mathrm{Gap}_{\mathcal{C}}(\bar{X}_T) = \mathcal{O}\left(\dfrac{H + 4M^3 + \log(1 + 4M^2 T)}{\sqrt{T}}\right).$  (9a)

*b) If $V$ satisfies* (LC)*:*   $\mathrm{Gap}_{\mathcal{C}}(\bar{X}_T) = \mathcal{O}(H/T).$  (9b)

Theorem 1 (which is proved in the sequel as a special case of Theorem 2) should be compared to the corresponding results of Bach & Levy [2]. In the non-smooth case, [2] provides a bound of the form $\tilde{\mathcal{O}}(\alpha M D/\sqrt{T})$ with $D^2 = \frac{1}{2}\max_{x\in\mathcal{X}} \|x\|^2 - \frac{1}{2}\min_{x\in\mathcal{X}} \|x\|^2$ (recall that [2] only treats problems with a bounded domain), and $\alpha = \max\{M/M_0, M_0/M\}$ where $M_0$ is an initial estimate of $M$. The worst-case value of $\alpha$ is $\mathcal{O}(M)$ when good estimates are not readily available; in this regard, (9a) essentially replaces the $\mathcal{O}(D)$ constant of Bach & Levy [2] by $\mathcal{O}(M)$. Since $D = \infty$ in problems with an unbounded domain, Theorem 1 provides a significant improvement in this regard.

In terms of $L$, the smooth guarantee of [2] is $\tilde{\mathcal{O}}(\alpha^2 L D^2/T)$, so the multiplicative constant in the bound also becomes infinite in problems with an unbounded domain. In our case, $D^2$ is replaced by $H$ (which is also finite) times an additional multiplicative constant which is increasing in $M$ and $L$ (but is otherwise asymptotic, so it is not included in the statement of Theorem 1). This removes an additional limitation in the results of [2]; in the next sections we drop even the Euclidean regularity requirements (BD)/(LC), and we provide a rate interpolation result that does not require either condition.

## 5   FINSLER REGULARITY

To motivate our analysis outside the setting of (BD)/(LC), consider the vector field

$$V_i(x) = (\mu_i - x_i)^{-1} + \lambda \mathbb{1}\{x_i > 0\}, \quad i = 1, \dots, N, \tag{10}$$

which corresponds to the distributed computing problem of Section 2.3 plus a regularization term designed to limit the activation of computing nodes at low loads. Clearly, we have $\|V(x)\| \to \infty$ whenever $x_i \to 0^+$, so (BD) and (LC) both fail (the latter even if $\lambda = 0$). On the other hand, if we consider the "local" norm $\|v\|_{x,*} = \sum_{i=1}^{d} (\mu_i - x_i)|v_i|$, we have $\|V(x)\|_{x,*} \le d + \lambda \sum_{i=1}^{d} \mu_i$, so $V$ is *bounded relative to* $\|\cdot\|_{x,*}$. This observation motivates the use of a *local* – as opposed to *global* – norm, which we define formally as follows:

**Definition 1.** A *Finsler metric* on a convex subset $\mathcal{X}$ of $\mathbb{R}^d$ is a continuous function $F: \mathcal{X} \times \mathbb{R}^d \to \mathbb{R}_+$ which satisfies the following properties for all $x \in \mathcal{X}$ and all $z, z' \in \mathbb{R}^d$:

1. *Subadditivity:* $F(x; z + z') \le F(x; z) + F(x; z').$
2. *Absolute homogeneity:* $F(x; \lambda z) = |\lambda| F(x; z)$ for all $\lambda \in \mathbb{R}$.
3. *Positive-definiteness:* $F(x; z) \ge 0$ with equality if and only if $z = 0$.

Given a Finsler metric on $\mathcal{X}$, the induced *primal/dual local norms* on $\mathcal{X}$ are respectively defined as

$$\|z\|_x = F(x; z) \quad \text{and} \quad \|v\|_{x,*} = \max\{\langle v, z \rangle : F(x; z) = 1\} \tag{11}$$

for all $x \in \mathcal{X}$ and all $z, v \in \mathbb{R}^d$. We will also say that a Finsler metric on $\mathcal{X}$ is *regular* when $\|v\|_{x',*}/\|v\|_{x,*} = 1 + \mathcal{O}(\|x' - x\|_x)$ for all $x, x' \in \mathcal{X}$, $v \in \mathbb{R}^d$. Finally, for simplicity, we will also assume in the sequel that $\|\cdot\|_x \geq \nu\|\cdot\|$ for some $\nu > 0$ and all $x \in \mathcal{X}$ (this last assumption is for convenience only, as the norm could be redefined to $\|\cdot\|_x \leftarrow \|\cdot\|_x + \nu\|\cdot\|$ without affecting our theoretical analysis).

When $\mathcal{X}$ is equipped with a regular Finsler metric as above, we will say that it is a *Finsler space*.

**Example 5.1.** Let $F(x; z) = \|z\|$ where $\|\cdot\|$ denotes the reference norm of $\mathcal{X} = \mathbb{R}^d$. Then the properties of Definition 1 are satisfied trivially. ◄

**Example 5.2.** For a more interesting example of a Finsler structure, consider the set $\mathcal{X} = (0, 1]^d$ and the metric $\|z\|_x = \max_i |z_i|/x_i$, $z \in \mathbb{R}^d$, $x \in \mathcal{X}$. In this case $\|v\|_{x,*} = \sum_{i=1}^d x_i |v_i|$ for all $v \in \mathbb{R}^d$, and the only property of Definition 1 that remains to be proved is that of regularity. To that end, we have

$$\|v\|_{x',*} - \|v\|_{x,*} \leq \sum_{i=1}^d |v_i| \cdot |x'_i - x_i| = \sum_{i=1}^d x_i |v_i| \cdot |x'_i - x_i|/x_i \leq \|v\|_{x,*} \cdot \|x' - x\|_x. \quad (12)$$

Hence, by dividing by $\|v\|_{x,*}$, we readily get $\|v\|_{x',*}/\|v\|_{x,*} \leq 1 + \|x - x'\|_x$ i.e., $\|\cdot\|_x$ is regular in the sense of Definition 1. As we discuss in the sequel, this metric plays an important role for distributed computing problems of the form presented in Section 2.3. ◄

With all this in hand, we will say that a vector field $V \colon \mathcal{X} \to \mathbb{R}^d$ is

1. *Metrically bounded* if there exists some $M > 0$ such that
$$\|V(x)\|_{x,*} \leq M \quad \text{for all } x \in \mathcal{X}. \quad \text{(MB)}$$

2. *Metrically smooth* if there exists some $L > 0$ such that
$$\|V(x') - V(x)\|_{x,*} \leq L\|x' - x\|_{x'} \quad \text{for all } x', x \in \mathcal{X}. \quad \text{(MS)}$$

The notion of metric boundedness/smoothness extends that of ordinary boundedness/Lipschitz continuity to a Finsler context; note also that, even though neither side of (MS) is unilaterally symmetric under the change $x \leftrightarrow x'$, the condition (MS) as a whole *is*. Our next example shows that this extension is *proper*, i.e., (BD)/(LC) may both fail while (MB)/(MS) both hold:

**Example 5.3.** Consider the change of variables $x_i \rightsquigarrow 1 - x_i/\mu_i$ in the resource allocation problem of Section 2.3. Then, writing $V_i(x) = -(1/x_i) - \lambda \mathbb{1}\{x_i < 1\}$ for the transformed field (10) under this change of variables, we readily get $V_i(x) \to -\infty$ as $x_i \to 0^+$; as a result, both (BD) and (LC) fail to hold for *any* global norm on $\mathbb{R}^d$. Instead, under the *local* norm $\|z\|_x = \max_i |z|_i/x_i$, we have:

1. For all $\lambda \geq 0$, $V$ satisfies (MB) with $M = d(1 + \lambda)$: $\|V(x)\|_{x,*} \leq \sum_{i=1}^d x_i \cdot (1/x_i + \lambda) = d(1 + \lambda)$.

2. For $\lambda = 0$, $V$ satisfies (MS) with $L = d$: indeed, for all $x, x' \in \mathcal{X}$, we have

$$\|V(x') - V(x)\|_{x,*} = \sum_{i=1}^d x_i \left| \frac{1}{x'_i} - \frac{1}{x_i} \right| = \sum_{i=1}^d \frac{|x'_i - x_i|}{x'_i} \leq d \max_i \frac{|x'_i - x_i|}{x'_i} = d\|x' - x\|_{x'}. \quad (13)$$

## 6 The AdaProx Algorithm and its Guarantees

**The method.** We are now in a position to define a family of algorithms that is capable of interpolating between the optimal smooth/non-smooth convergence rates for solving (VI) without requiring either (BD) or (LC). To do so, the key steps in our approach will be to (*i*) equip $\mathcal{X}$ with a suitable Finsler structure (as in Section 5); and (*ii*) replace the Euclidean projection in (EG) with a suitable "Bregman proximal" step that is compatible with the chosen Finsler structure on $\mathcal{X}$. We begin with the latter (assuming that $\mathcal{X}$ is equipped with an arbitrary Finsler structure):

**Definition 2.** We say that $h : \mathbb{R}^d \to \mathbb{R} \cup \{\infty\}$ is a *Bregman-Finsler function* on $\mathcal{X}$ if:

1. $h$ is convex, lower semi-continuous (l.s.c.), $\mathrm{cl}(\mathrm{dom}\, h) = \mathrm{cl}(\mathcal{X})$, and $\mathrm{dom}\, \partial h = \mathcal{X}$.
2. The subdifferential of $h$ admits a *continuous selection* $\nabla h(x) \in \partial h(x)$ for all $x \in \mathcal{X}$.
3. $h$ is *strongly convex*, i.e., there exists some $K > 0$ such that
$$h(x') \geq h(x) + \langle \nabla h(x), x' - x \rangle + \frac{K}{2}\|x' - x\|_x^2 \quad (14)$$
for all $x \in \mathcal{X}$ and all $x' \in \mathrm{dom}\, h$.

The *Bregman divergence* induced by $h$ is defined for all $x \in \mathcal{X}$, $x' \in \text{dom } h$ as

$$D(x', x) = h(x') - h(x) - \langle \nabla h(x), x' - x \rangle \tag{15}$$

and the associated *prox-mapping* is defined for all $x \in \mathcal{X}$ and $y \in \mathbb{R}^d$ as

$$P_x(y) = \arg\min_{x' \in \mathcal{X}} \{\langle y, x - x' \rangle + D(x', x)\}. \tag{16}$$

Definition 2 is fairly technical, so some clarifications are in order. First, to connect this definition with the Euclidean setup of Section 4, the prox-mapping (16) should be seen as the Bregman equivalent of a Euclidean projection step, i.e., $\Pi(x+y) \leftrightsquigarrow P_x(y)$. Second, a key difference between Definition 2 and other definitions of Bregman functions in the literature [4, 6, 7, 9, 24, 36, 37, 46] is that $h$ is assumed strongly convex relative to a *local* norm – not a global norm. This "locality" will play a crucial role in allowing the proposed methods to adapt to the geometry of the problem. For concreteness, we provide below an example that expands further on Examples 5.2 and 5.3:

**Example 6.1.** Consider the local norm $\|z\|_x = \max_i |z_i|/x_i$ on $\mathcal{X} = (0, 1]^d$ and let $h(x) = \sum_{i=1}^d 1/x_i$ on $(0, 1]^d$. We then have

$$D(x', x) = \sum_{i=1}^d \left[ \frac{1}{x_i'} - \frac{1}{x_i} + \frac{x_i' - x_i}{x_i^2} \right] = \sum_{i=1}^d \frac{(x_i' - x_i)^2}{x_i^2 x_i'} \geq \sum_{i=1}^d (1 - x_i'/x_i)^2 \geq \|x' - x\|_x^2 \tag{17}$$

i.e., $h$ is 1-strongly convex relative to $\|\cdot\|_x$ on $\mathcal{X}$.  ◄

With all this is in place, the extra-gradient method can be adapted to our current setting as follows:

$$
\begin{aligned}
X_{t+1/2} &= P_{X_t}(-\gamma_t V_t) & \delta_t &= \|V_{t+1/2} - V_t\|_{X_{t+1/2},*} \\
X_{t+1} &= P_{X_t}(-\gamma_t V_{t+1/2}) & \gamma_{t+1} &= 1 \Big/ \sqrt{1 + \sum_{s=1}^t \delta_s^2}
\end{aligned}
\tag{AdaProx}
$$

with $V_t = V(X_t)$, $t = 1, 3/2, \ldots$, as in Section 3. In words, this method builds on the template of (EG) by (*i*) replacing the Euclidean projection with a mirror step; (*ii*) replacing the global norm in (8) with a dual Finsler norm evaluated at the algorithm's leading state $X_{t+1/2}$.

**Convergence speed.** With all this in hand, our main result for AdaProx can be stated as follows:

**Theorem 2.** *Suppose $V$ satisfies* (Mon)*, let $\mathcal{C}$ be a compact neighborhood of a solution of* (VI)*, and set $H = \sup_{x \in \mathcal{C}} D(x, X_1)$ Then, the AdaProx algorithm enjoys the guarantees:*

*a) If $V$ satisfies* (MB)*:*  $$\text{Gap}_{\mathcal{C}}(\bar{X}_T) = \mathcal{O}\left( \frac{H + M^3(1 + 1/K)^2 + \log(1 + 4M^2(1 + 2/K)^2 T)}{\sqrt{T}} \right). \tag{18a}$$

*b) If $V$ satisfies* (MS)*:*  $$\text{Gap}_{\mathcal{C}}(\bar{X}_T) = \mathcal{O}(H/T). \tag{18b}$$

For the constants that appear in Eq. (18), we refer the reader to the discussion following Theorem 1. Moreover, we defer the proof of Theorem 2 to the paper's supplement. We only mention here that its key element is the determination of the asymptotic behavior of the adaptive step-size policy $\gamma_t$ in the non-smooth and smooth regimes, i.e., under (MB) and (MS) respectively. At a very high level, (MB) guarantees that the difference sequence $\delta_t$ is bounded, which implies in turn that $\sum_{t=1}^T \gamma_t = \Omega(\sqrt{T})$ and eventually yields the bound (18a) for the algorithm's ergodic average $\bar{X}_T$. On the other hand, if (MS) kicks in, we have the following finer result:

**Lemma 1.** *Assume $V$ satisfies* (MS)*. Then, a) $\gamma_t$ decreases monotonically to a strictly positive limit $\gamma_\infty = \lim_{t\to\infty} \gamma_t > 0$; and b) the sequence $\delta_t$ is square summable: in particular, $\sum_{t=1}^\infty \delta_t^2 = 1/\gamma_\infty^2 - 1$.*

By means of this lemma (which we prove in the paper's supplement), it follows that $\sum_{t=1}^T \gamma_t \geq \gamma_\infty T = \Omega(T)$; hence it ultimately follows that AdaProx enjoys an $\mathcal{O}(1/T)$ rate of convergence under (MS).

**Trajectory convergence.** In complement to Theorem 2, we also provide a trajectory convergence result that governs the *actual* iterates of the AdaProx algorithm:

**Theorem 3.** *Suppose that $\langle V(x), x - x^* \rangle < 0$ whenever $x^*$ is a solution of* (VI) *and $x$ is not. If, in addition, $V$ satisfies* (MB) *or* (MS)*, the iterates $X_t$ of AdaProx converge to a solution of* (VI)*.*

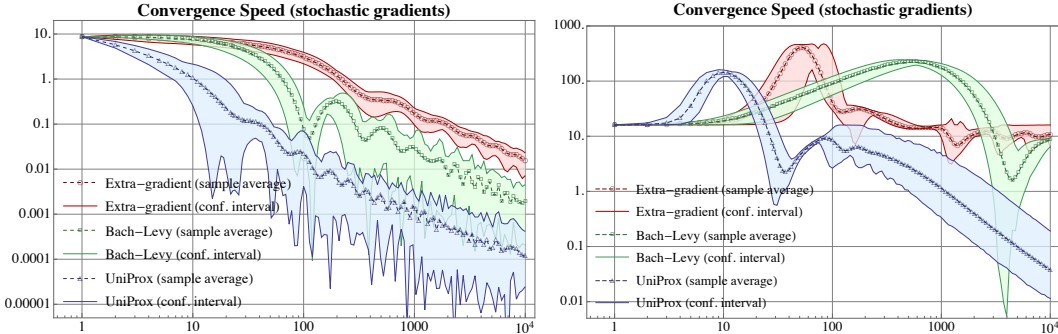

**Figure 2:** Numerical comparison between the extra-gradient (EG), Bach–Levy (BL) and AdaProx algorithms (red circles, green squares and blue triangles respectively). The figure on the left shows the methods' convergence in a $100 \times 100$ bilinear game; the one on the right shows the methods' convergence in a non-convex/non-concave covariance learning problem. In both cases, the parameters of the EG and BL algorithms have been tuned with a grid search (AdaProx has no parameters to tune). All curves have been averaged over $S = 100$ sample runs, and the 95% confidence interval is indicated by the shaded area.

The importance of this result is that, in many practical applications (especially in non-monotone problems), it is more common to harvest the "last iterate" of the method $(X_t)$ rather than its ergodic average $(\bar{X}_T)$; as such, Theorem 3 provides a certain justification for this design choice.

The proof of Theorem 3 relies on non-standard arguments, so we relegate it to the supplement. Structurally, the first step is to show that $X_t$ visits any neighborhood of a solution point $x^* \in \mathcal{X}^*$ infinitely often (this is where the coherence assumption $\langle V(x), x - x^* \rangle$ is used). The second is to use this trapping property in conjunction with a suitable "energy inequality" to establish convergence via the use of a quasi-Fejér technique as in [10]; this part is detailed in a separate appendix.

## 7 NUMERICAL EXPERIMENTS

We conclude in this section with a numerical illustration of the convergence properties of AdaProx in two different settings: *a)* bilinear min-max games; and *b)* a simple Wasserstein GAN in the spirit of Daskalakis et al. [12] with the aim of learning an unknown covariance matrix.

**Bilinear min-max games.** For our first set of experiments, we consider a min-max game of the form of the form $\mathcal{L}(\theta, \phi) = (\theta - \theta^*)^\top A(\phi - \phi^*)$ with $\theta, \phi \in \mathbb{R}^{100}$ and $A \in \mathbb{R}^{100} \times \mathbb{R}^{100}$ (drawn i.i.d. component-wise from a standard Gaussian). To test the convergence of AdaProx beyond the "full gradient" framework, we ran the algorithm with stochastic gradient signals of the form $V_t = V(X_t) + U_t$ where $U_t$ is drawn i.i.d. from a centered Gaussian distribution with unit covariance matrix. We then plotted in Fig. 2 the squared gradient norm $\|V(\bar{X}_T)\|^2$ of the method's ergodic average $\bar{X}_T$ after $T$ iterations (so values closer to zero are better). For benchmarking purposes, we also ran the extra-gradient (EG) and Bach–Levy (BL) algorithms [2] with the same random seed for the simulated gradient noise. The step-size parameter of the EG algorithm was chosen as $\gamma_t = 0.025/\sqrt{t}$, whereas the BL algorithm was run with diameter and gradient bound estimation parameters $D_0 = .5$ and $M_0 = 2.5$ respectively (both determined after a hyper-parameter search since the only *theoretically* allowable values are $D_0 = M_0 = \infty$; interestingly, very large values for $D_0$ and $M_0$ did not yield good results). The experiment was repeated $S = 100$ times, and AdaProx gave consistently faster rates.

**Covariance matrix learning.** Going a step further, consider the covariance learning game

$$\mathcal{L}(\theta, \phi) = \mathbb{E}_{x \sim \mathcal{N}(0, \Sigma)}[x^\top \theta x] - \mathbb{E}_{z \sim \mathcal{N}(0, I)}[z^\top \theta^\top \phi \theta z], \qquad \theta, \phi \in \mathbb{R}^d \times \mathbb{R}^d. \tag{19}$$

The goal here is to generate data drawn from a centered Gaussian distribution with unknown covariance $\Sigma$; in particular, this model follows the Wasserstein GAN formulation of Daskalakis et al. [12] with generator and discriminator respectively given by $G(z) = \theta z$ and $D(x) = x^\top \phi x$ (no clipping). For the experiments, we took $d = 100$, a mini-batch of $m = 128$ samples per update, and we ran the EG, BL and AdaProx algorithms as above, tracing the square norm of $V$ as a measure of convergence. Since the problem is non-monotone, there are several disjoint equilibrium components so the algorithms' behavior is considerably more erratic; however, after this initial warm-up phase, AdaProx again gave the faster convergence rates.

## ACKNOWLEDGMENTS

This research was partially supported by the COST Action CA16228 "European Network for Game Theory" (GAMENET) and the French National Research Agency (ANR) in the framework of the grants ORACLESS (ANR–16–CE33–0004–01) and ELIOT (ANR-18-CE40-0030 and FAPESP 2018/12579-7), the "Investissements d'avenir" program (ANR-15-IDEX-02), the LabEx PERSYVAL (ANR-11-LABX-0025-01), and MIAI@Grenoble Alpes (ANR-19-P3IA-0003).

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
