# OpenReview forum: "Adaptive Extra-Gradient Methods for Min-Max Optimization and Games"
_ICLR.cc/2021/Conference — ICLR 2021 Poster_

### Official Review · AnonReviewer1 · 2020-10-20

**Rating:** 7
**Confidence:** 4

**Review:**

In this paper, the authors present an adaptive stepsize strategy for extragradient in order to recover both the ergodic 1/sqrt(T) in the nonsmooth case and 1/T in the smooth case (without the knowledge of the Lipchitz constant). The presented method is rather simple but seems effective both in theory and in practice.

The paper is well written: the motivation, context, and basic definitions are especially clear. The algorithm is intersting and the analysis seems sound. I'm in favor of acceptance.

Concerns
---------

* Could the authors comment on X in 2.3? To be compatible with the projection as used in the EG, it probably has to be closed and known in advance. With the inclusion between K and its relative interior, this seems to imply that K has to be closed somehow.

* The auhors mention that their stepsize strategy adapts to local Lipschitz constants but since the points are uniformly averaged since iteration 1, the local geometry may be a bit lost. Would it be possible to do some kind of restarting to mitigate this effect?

* I have troubles understanding the statemeent of Theorem 2, notably what does gap(*)<0 means or what are the implication of C being relatively closed and disjoint.



Minor Comments:
- First paragraph: 9 citations in a single pair of brackets is not so informative, please consider breaking it down to smaller bits organized bu great topics. (Same remark at the beginning of Section 3).
- Caption of Table 1: algortihm -> algorithm
- In related works: the notions of "interpolation" (as defined in the caption of Tab.1) and "divergent operators" (in the sense of coercive at the boundary?) may not be so clear to the reader. An additional explanation in the text or a footnote would be appreciated.
- Below (8), I think the authors mean "lim inf \delta_t >0" and not "lim inf \gamma_t >0"
- The examples sometime end with a small triangle, sometimes not.
- It is a small detail but gamma_t is defined in (8) but gamma_{t+1} is defined in (UniProx), this could be made uniform.
- Could the authors comment on the computability of the proximal mapping (16)? Since X may not be closed, there may be problems at the border. This is probably handled by h as in the example but I am not sure it is implied by the conditions (eg the stong convexity in the local metric).
- Fig. 2 right appears poorly when printed.
- I would have appreciated a last-iterate vs Average iterate behavior comparison as well as locally smooth/nonsmooth examples in the experiments. (For instance, instead of the stochastic case which seems a bit off topic).
- Parts of the proof in the appendix could be more detailed:
-- Precise that you use regularity in C.7
-- When you use Fenchel-Young, you introduce a K without explicitly mentioning it (eg D.8 or D.22).
-- Precise the derivations between D.26 and D.27 (precise that Lemma C.1 is used)

---

> ### Author Response · Authors · 2020-11-18
> **Replies to AnonReviewer1**
>
> We are sincerely grateful to the reviewer for their positive evaluation and detailed input. We are very happy that the reviewer enjoyed our paper, and we reply to their discussion points below.
>
> 1. On the players' action sets $\mathcal{K}_i$: we do not necessarily assume that these sets are closed. This is for the following reason: if we want to minimize the function $f(x) = \log x + \log(1-x)$ over $(0,1)$, we cannot take a closed domain because of the singularities at the endpoints. The extra-gradient algorithm is not well-equipped to handle problems with such a domain, because the Euclidean regularizer on the open interval $(0,1)$ is not lower semi-continuous (cf. Definition 2) so $\mathcal{K}$ must indeed be closed in the case of EG. This distinction was not expressed with sufficient clarity, so we took this opportunity to add a number of relevant remarks at several points in the paper (when introducing $\mathcal{K}$, when defining the extra-gradient algorithm, in and around Definition 2, etc.).
>
> 2. On restarting the algorihtm: this is a very nice idea! Since our derivations are "regret-based" (in the language of online learning), we believe that a restart mechanism coupled with a suitable doubling trick should be able to achieve what the reviewer suggests (and also achieve better intra-class adaptation properties, as we mentioned to AnonReviewer5). However, the calculations and the entire approach would be considerably different in this case, so we cannot pursue this approach in the present paper.
>
> 3. Regarding Theorem 2 (Theorem 3 in our revision): this is a common, Minty-like assumption. We changed the formulation to make it clearer, thanks for pointing out that it was not clear in the original version of our paper.
>
> **Minor points:**
> 1. Cluster of references in the first paragraph: fixed.
> 2. Typo in Table 1: fixed.
> 3. For Table 1: the notion of "rate interpolation" is defined clearly in the caption. The terminology "divergent" has been abandoned in favor of "singularities", and we also broke up the unbounded domain / singularity lines to make things even clearer.
> 4. Typo below (8): fixed, thanks!
> 5. End of examples: fixed when possible, thanks.
> 6. $\gamma_t$ vs. $\gamma_{t+1}$ – sure thing, done.
> 7. Indeed, strong convexity implies coercivity, and the fact that $\mathrm{dom} \partial h = \mathcal{X}$ implies that the minimum is attained at a point in $\mathcal{X}$.
> 8. Fig. 2: we increased the resolution of the figure. In our printouts, the figure looks fine, but we'll be happy to investigate further if the problem persists.
> 9. The stochastic case is now more relevant because of the other reviewers' urging to consider experiments with GANs and/or other stochastic problems. We are not opposed to adding more experiments if the reviewer insists, but this is a theory paper, so we would like to maintain the focus on our theoretical results.
> 10. Points regarding the supplement: thank you for pointing these out, they will all be fixed.
>
> Again, our sincere thanks for your time and efforts, we are greatly obliged for both.

---

> > ### Comment · AnonReviewer1 · 2020-11-20
> > **Quick comment on the response**
> >
> > I am satisfied with the authors' response. I have a few very minor comments:
> > * "Suppose V satisfies (MC)" in Theorem 3 for instance feels a bit unnecessarily involved way of saying "Suppose that V is monotonous". Since there are already several 2-letters acronyms, dropping this one could improve readability.
> > * In the same train of though, in  BD vs MB , the important word is bounded(B), hence BD and BM could be more easily understood; same thing for LC vs MS , here metrically smooth seems a bit confusing since it usually refers to Lipschitz derivatives... But all that may be a question of taste.
> > * In "trajectory convergence", when you say "the actual iterate", you probably mean the non-averaged ones (I was confused for a moment)

---

> > > ### Author Response · Authors · 2020-11-24
> > > **Thanks for the remarks**
> > >
> > > Thank you for this last round of remarks, we commit to take them into account (we did not provide a second revision of the pdf to avoid overloading the thread in case other reviewer replies also came through at the same time).

---

### Official Review · AnonReviewer4 · 2020-10-27
**Review on 'Adaptive Extra-Gradient Methods for Min-Max Optimization and Games'**

**Rating:** 7
**Confidence:** 4

**Review:**

In this paper the authors propose a variant of the extragradient method that i) achieves optimal convergence in the smooth and nonsmooth regimes, ii) allows one to work with singular operators. The paper is written very well and as far as I could check, mathematics is correct (apart from minor details which are easy to fix). Most importantly, in their analysis the authors use a new tool to work with unbounded operators, which may be of interest in its own right.


Although I applaud authors for the use of innovative tools in their analysis, I would like to raise some concerns for a possible discussion.  Mostly, I am not so happy with the practical side of the paper:

1) No concrete important problems intractable before by the standard extragradient but tractable by the proposed version.

2) Quite bad constants. I think at some point the authors forgot one extra $M$ (see my detailed comment below), but even without it the rates are good asymptotically but bad if constants are written explicitly. I like that the authors did provide explicit formulas, but still feel that some discussion about it might be appropriate. Also I think it would help to have explicit constants in the (MS) case, though I understand that it is harder.
For example, no doubts that the rate interpolation is a great property. On the other hand, for the extragradient choosing right steps between $1/L$ or $1/\sqrt{t}$ does not seem to be so difficult, especially if this choice will appear to be much faster in practice.

3) Discussion of how to choose a Finsler metric is missing (apart from simple examples). For example, what if $V$ is well-defined but not Lipschitz, like $V(x) = \exp(x)$? It is not Lipschitz, so looking at Table 1, a reader can think that this case is covered by the proposed analysis. However, it will be cowered only when a suitable $F$ is given, which (i) a non-trivial problem by itself, (ii) probably not even achievable in principle for all such $V$. In other words, can we prove that for any such $V$ described above there exist a suitable $F$? I doubt it, since an unbounded domain must matter here.

4) I feel like the authors unconsciously push me to suspect that the proposed algorithm will not perform well in most cases. First, as I said above, large constants in the rate and omitted discussion of it. Second, not so illuminating numerical results (Figure 2). I believe the algorithm was designed not for solving unconstrained minimax problem. This problem is too easy for any solver (and to be fair, the comparison with other algorithms was quite limited). Where are some demonstrations of its performance for hard problems (with singular operator)? If not a real world problem, then at least the ones that motivated this paper. Not that even Example 5.3 is not very interesting, since for $V(x) = -(1/x_1, \dots)$ and $X=(0,1]$ the solution will be $x=(1,\dots)$, thus the singularity at zero unlikely will play any role. For instance, what about convex minimization with barriers $\min f(x) - \log x$ on $(0,1]$?

Independently of this, I am in favor of proposed analysis.  I think theoretical contribution is much more important here and it is OK for now if a practical performance still lacks behind. However, it would be nice and helpful to warn readers about possible obstacles and also to motivate them for further research.


Below I list some issues I found. Most of them are minor, but some not.


1. page 1 "Originally applied to variational inequalities": I may be wrong, but I think originally it was applied to saddle point problems, though the extension is of course mostly a change of notation.

2. page 1 "if coupled with an iterate averaging scheme -- the mirror-prox...": Not sure what the authors mean. The mirror-prox template is not needed for getting an ergodic $O(1/T)$ rate. Why do we need mirror updates for that?

3. page 2, Contribution #3 "Off-the-shelf": I cannot entirely agree with it. While this is definitely true for rate interpolation, for addressing Lipschitz continuity this is not true. The proposed algorithm still requires a Bregman-Finsler function as an input. In other words, it requires a user to find such a function and verify one of conditions (MB) or (MS). The user just does not need a precise values of $M$ or $L$, which is good of course.

4. page 2 "As an extra feature of our analysis, we also show that the sequence of iterates...": I would not call this as an extra feature, usually it is one of the essential algorithm's attributes in continuous optimization. Moreover, it is not entirely true, since it requires another assumption that is impossible to verify in practice.

5. page 3: I think adding  "there exists $i$" or "for all $i$"  will help to make a definition of smooth/nonsmooth games more clear.

6. page 3, Proposition 1: its formulation is ambiguous at the moment. Should this inequality hold for any collection of vector fields $V$ or is it sufficient to have only one?

7. Eq. (8):  1 in the denominator. How sensitive is it? Obviously, it should matter for the convergence in the end.
8. L is not present in Eq. (9b)

9. page 6: How standard is the given definition of a regular Finsler metric? It is worth to add the reference, if it is standard.

10. page 6 "The KL loss is well-defined..." I didn't understand this sentence. How is it defined when $x = 0$?

11. page 6 "and the only axiom left to verify is..." I got an impression from above that regularity is not an axiom, but just and an additional property that is good to have.

12. page 6 "the norm could be redefined to ... without affecting our results": I am not sure how valid is this remark. First, we should distinguish theoretical results and practical. For theory I agree, though it was needed only for Theorem 2, not Theorem 3, which possibly may be highlighted. For practical purposes though it might be undesirable to change the norm, since we want $h$ to be strongly convex w.r.t. this norm. And if we start to change $h$, then our algorithmic update will be more expensive, etc. Also it is not clear whether $V$ will still satisfy (MB) or (MS): the dual norm will be also different.

13. page 6 "Metrically smooth" definitions is not clear: it is not symmetric to $x, x'$. Maybe some discussion?

14. page 6, Example 5.3: $V$ is not monotone, thus it contradicts to the "blanket assumption". Considering $V$ with $-$ will be better.

15. page 7 "Second, the main difference between Definition 2 and other definitions of Bregman proximal mappings": I think the authors meant Bregman function, the definition of the proximal mapping is standard.

16. I think it is better to mention all assumptions in the statements (even just as abbreviation or references). Currently, they are put everywhere in the paper. For a reader who wants to quickly skim the main results but will not read a full paper it will be quite helpful.

17. page 7, Th.2: Is set $C$ is fixed or not? The same for a solution $x^*$. And of course the condition is a bit strange.

18. Why not to swap Th.2 and 3? The latter is more important and doesn't require an additional assumption. The former looks more like a side result.

19. page 8, Figure 2: What does it mean tuned or untuned for Extragradient? Larger step? Why do we care about it, isn't it obvious that the algorithm should diverge? Why not to take $\gamma =\frac{1}{|A|}$? How $A$ was generated? It is worth to have a better accuracy for plots in right picture. How the problems in right were generated is not clear. If $V$ was contaminated by noise, does it imply that $V$ is not monotone?

20. page 8, "... Fig. 2 confirms that UniProx can provide a fruitful template for adaptive min-max optimization methods that attain optimal rates in smooth/non-smooth problems." Two plots for a smooth problem unlikely can confirm a smooth case and certainly not a non-smooth one.

21. page 12, Lemma B.1. Note that there wasn't given a definition of Bregman function, but only Bregman-Finsler one.

22. page 12, " if and only if $t=0$": why? what if $p=x$. Next sentence: $\psi$ is a function and not a continuous selection. Next: what does it mean for $\phi$ to be differentiable at $t=0$ or $t=1$?

23. page 13: I don't think notation $X^\circ$ was introduced before. What is a local Bregman function?

24. page 13: I did not understand how condition (C.2) is more general. Note that the definition of regularity uses asymptotic language, that is when $|x'-x|_x \to \infty$. The condition (C.2) is global on the other hand. Of course, there is some intersection of two conditions, but not in general. Maybe instead of regularity, the authors should consider (C.2) definition from the very beginning.

25. Lemma C.1, page 14. I have some difficulties with the proof of this lemma. First, I didn't understand the first sentence: why changing the reference point from $X_t$ to $X_{t+1/2}$ will be sufficient for us? Second, I did not understand the derivation (C.7). Apparently the authors use condition (C.2) but they forgot an extra factor in the RHS, i.e, there should be $M$ next to $\beta$. This will worsen the final bound of course.

26. Lemma D.1: In the main part the authors use notation $\gamma_\infty$ instead of $\alpha$.

27. page 16, Eq.(D.12): it should be $\gamma_{T+1}^2$ in the LHS.

28. I think that parts of the proof of Lemma D.1 can be reused in the proof of Theorem 3.

29. Eq. (D.22): the RHS should be divided over $\sum \gamma_t$.

30. page 18, "Hence, $\gamma_t$ is non-increasing...". This fact is definitely not a consequence of the equation above. Also the equation below should have $\gamma_{t+1}^2$.

31. page 18, "For the second term...": that term should be have a dual norm (star is missing). Also everywhere below.

32. page 18: It makes sense to recall what $C$ is. Note that (C.13) for this is good, but (C.4) is not, since it uses a different reference point. Do we actually need (C.4)? Also probably having a set $\mathcal{C}$ and a constant $C$ in the same proof is not the best notation choice.

33. page 18, "we have the following upper-bound": lower.

34. page 18, Eq. (D.30): Note that in the original statement of Th.3 there were exact constants in $O()$.

35. page 18, Eq.(D.31): Why do we need this equation? I think we should apply Lemma D.1 to Eq.(D.24). Otherwise it is not clear how the bound for the gap follows.

36. page 19: I didn't understand (Eq.D.32). Why $\mathrm{dom} (V) = \mathrm{dom} (h)$ and why not just use the same compactness argument as above? Also, it should be $p$ and not $x^*$.

37. page 19, (Eq.D.34): $\gamma_t^2$ is redundant.

38. page 20, (Eq.E.4): $t$ and $j$ are mixed.

39. page 20, Eq. E.8-E.10: Almost the same things are already proven in Lemma C.1. The explanations there are also more transparent.

40. page 21: Wrong reference in Remark 2.

41. page 21: Proposition E.1: Why now equilibrium set and not a solution set?

42. page 21: I think both in the statement of Prop E.1 and in Eq.(E.19) the extra condition from Th.2 should be mention. Otherwise I don't understand how E.19 was obtained.

43. page 21: Summation in Eq.(E.20) should start not from $t=1$. Liminf in E.19 means that that inequality holds for all $t$ starting from $t=t_0$.

---

> ### Author Response · Authors · 2020-11-18
> **Reply to major points**
>
> We are deeply grateful to the reviewer for their extremely thorough and insightful review – it was one of the most detailed reviews we have ever received in an ML conference. We are very happy for the reviewer's positive evaluation, and we have uploaded a drastically revised version of our paper to take into account these remarks. For the reviewers' convenience, we have highlighted all relevant changes in blue, and we also reply below to the reviewer's major discussion points.
>
> **1. Concrete problems intractable by extra-gradient.** We combined this with a point raised by AnonReviewer5, and we now provide a concrete running example taken from distributed grid computing. Because of the problem's $1/x$ singularity at the boundary of its domain, the problem cannot be tackled by standard EG methods, and it requires the full capacity of our framework. This running example forms the core of Section 2.3, and the examples of Finsler metrics and regularizers have also been tuned accordingly.
>
> **2. Suboptimal constants.** First off, indeed, there was a factor of $M$ missing in (9a) and (18a), many thanks for pointing this out! On the other hand, we are not sure why the reviewer sees these constants as suboptimal: for example, in the non-smooth case, the corresponding constant in the work of Bach & Levy (2019) is $\alpha M D$ where $D$ is the (Bregman) diameter of the problem's domain and $\alpha = \max\{M/M_0,M_0/M\}$ with $M_0$ being an initial guess for $M$. If there is no prior information, the worst-case value of $\alpha$ is $O(M)$ so, in this regard, our bound *reduces* the Bach-Levy multiplicative constant from $M^2 D$ (which is infinite in problems with an unbounded domain) to $M^3$. We now provide a detailed discussion of this issue in Section 4, right after Theorem 1.
>
> **3. Choice of Finsler metric.** For this, it suffices to have an idea of the type of singularities that may develop at the boundary of the problem's domain. To make this precise, suppose that one wants to apply AdaProx to solve a Poisson inverse problem. Even if nothing else is known about the specific instance of the problem (i.e., before receiving the problem's observed data matrix), it is already known that Poisson objectives exhibit logarithmic singularities, so choosing a Finsler metric with a $1/x$ behavior at the boundary of the problem's domain suffices.
>
> In the reviewer's specific example, it would suffice to take the local norm $\vert z\vert_x = exp(x) \vert z\vert$ : the corresponding dual norm is $exp(-x) \|z\|$ so $V(x)$ satisfies (MB) with $M=1$. More generally, the basic rule of thumb is: if $V$ exhibits an $O(f(x))$ singularity at some point, choose a Finsler metric that grows like $\Theta(f(x))$ near the singularity. [More concretely, $V$ is always metrically bounded under the Finsler norm $\|z\|_x = \|z\| \times (1 + \|V(x)\|)$.]   We hope this clarifies how a Finsler metric can be chosen in a systematic way.
>
> **4. Constants and numerical results.** First, for the constants, please see above: mutatis mutandis, the constants that appear in the analysis of AdaProx are directly comparable to those of Bach & Levy (2019), so we do not see why they are suboptimal in this regard. Concerning more difficult problems, and given the other reviewers' urging to provide a connection to GANs, we now present an experiment on the Wasserstein GAN model presented in the ICLR paper of Daskalakis et al (2018) for learning covariance matrices. This model is non-convex/non-concave, and contains many disjoint equilibrium components, so it is quite difficult to solve, even in low dimensions. Nevertheless, the results that we observe are similar to the bilinear case, with AdaProx achieving the faster convergence rates. Of course, as the reviewer notes later, this does not *confirm* anything per se (we removed that passage), but we nonetheless believe that it is strongly encouraging evidence in support of the theory developed in our paper.
>
> In our revision, we implemented the above changes in Section 2.3 (which is new), the discussion after Theorem 1 in Section 4, we rewrote a good part of Sections 5 and 6, and we added Section 7 to present our numerical results in more detail. We also included the reviewer's minor suggestions in the main paper, and we will do the same for the reviewer's minor comments in the supplement as well.
>
> To keep the discussion flowing, we reply to the reviewer's minor remarks in a separate message. Needless to say, we remain at the reviewer's disposal for any clarifications, and we thank them again for their deep review and positive recommendation.

---

> ### Author Response · Authors · 2020-11-18
> **Reply to minor points**
>
> In this second part of our reply, we address the reviewer's minor points one-by-one - and we repeat our thanks for their thorough review.
>
> **TLDR:** All points concerning the main paper have already been implemented in the revision version uploaded to OpenReview; all other minor points concerning the supplement will be likewise addressed once the discussion phase has been concluded.
>
> We now proceed to the reviewer's remarks, following the enumeration in their original review (please note that some page and theorem numbers have changed in our revision).
>
> **Minor remarks concerning the main paper:**
>
> 1. EG for VIs: fixed (it's hard to tell from the original text in Russian, the VI formulation referred to the "other problems" in the paper's title).
> 2. Averaging: we now refer to Rupper-Polyak averaging instead.
> 3. "Off-the-shelf": removed throughout the paper.
> 4. Last iterate as "extra feature": we changed the order of presentation and the formulation to clarify (note that Theorems 2 and 3 have changed enumeration as a result of this change).
> 5. Player index: added.
> 6. Proposition 1: it has been removed. As all such selections are subgradients, we don't find the current formulation ambiguous but we'd be happy to add more details if needed.
> 7. This ties in to a remark of AnonReviewer5: order-optimality is achieved with any constant instead of $1$ though, of course, the explicit expressions will be affected. The role of this choice would be roughly analogous to the initial guess $M_0$ in the Bach-Levy algorithm; we did not include it because our focus was order-optimality, not the explicit constants.
> 8. Please see the detailed discussion right after Theorem 1.
> 9. We are not aware of a specific reference – we rephrased accordingly.
> 10. The presentation and focus has changed so this point is no longer relevant. [But, for completeness, it is defined via the convention $0 \log 0 = 0$.]
> 11. Done, changed "axiom" to "property".
> 12. If the primal norm grows, the dual norm diminishes, so (MB) and (MS) are still guaranteed in this case. This could also be relaxed by asking that $h$ be coercive, but we found that this assumption is the simplest one in terms of presentation.
> 13. Note that, even though each side of (MS) is not individually symmetric with respect to the change $x \leftrightarrow x'$, the condition as a whole *is*. We added a remark to this effect right after (MS).
> 14. Whoops... fixed, thanks!
> 15. Corrected the statement, thanks.
> 16. We updated all theorem statements accordingly.
> 17. We changed the formulation to make things clearer.
> 18. Theorems 2 and 3: swapped.
> 19. Fig. 2 has been entirely reworked.
> 20. Changed the formulation.
>
> **Minor remarks concerning the supplement:**
>
> 21. Will fix.
> 22. If $p=x$, there is nothing to prove (we will clarify this). Differentiability means one-sided at the endpoints of $[0,1]$. Finally, we are not sure why $\psi$ is not a selection of subgradients of $\phi$ (since $\phi$ is a function of a one-dimensional variable).
> 23. The $^\circ$ was a typo and local Bregman should be "Bregman-Finsler".
> 24. We will correct the formulation, thanks.
> 25. The reference point was a typo, our apologies (it should be $t+1/2$ throughout). The reviewer is also correct about the missing $M$ factor; we have implemented this in the main paper, and will do so in the supplement as well.
>
> 26 through 29. Yes, will fix.
>
> 30. The "hence" should be a "since".
>
> 31 through 43. Will fix - many thanks for spotting these!
>
> Again, we are deeply grateful to the reviewer for their extremely detailed and thorough input, we sincerely appreciate their time and efforts.

---

### Official Review · AnonReviewer3 · 2020-11-05
**Interesting aspect, more experiments needed**

**Rating:** 6
**Confidence:** 2

**Review:**

This paper propose a novel algorithm that solves the min-max problems and games based on the extra gradient (EG) framework. One of the main goal of this paper is to achieve the optimal convergence rate for both smooth/nonsmooth setttings without assuming the Lipchitz continuity/boundedness conditions. A "Bregman-Proximal" step was introduced to take place of the traditional Euclidean norm projection norm to depict the geometry or the smoothness properties of the problem. Furthermore, the authors adopt an adaptive step size scheme into the "Mirror-Prox" step to achieve the optimal convergence rate under both settings.

Strengths:
+ Achieves the optimal convergence rate in both the smooth case without the Lipchitz continuity assumptions ($O(1/T)$) and the non-smooth case without the boundedness assumption ($O(1/\sqrt{T})$)
+ "Off-the-shelf" design frees the users from tuning parameters
+ The main improvement over previous Bregmen Mirror Prox algorithms (K Antonakopoulos 2019) is the novel nonsmooth analysis under the MB condition

Weaknesses:
- The authos made an effort to show evaluation results on a basic synthetic example, but this is less challenging.
- Some of the details in the experimental sections are missing, for example the choice of the matrix A.
- The analysis of the convergence rate of a mirror step under the MB condition is an application of the techniques in AMP

In summary, the UnixProx algorithm achieves the optimal $1/T$ convergence or $1/\sqrt{T}$ convergence under the smooth/nonsmooth settings, while it does not require Lipchitz/Boundedness assumptions, which is applicable to problems without prior knowledge of the objective smoothness. This work can potentially help in solving games and reinforcement learning problems. However, the authors are encouraged to provide more experimental results to illustrate the practical impact of relaxing the assumptions.

Question:
can the convergence result in Theorem 1 be interpreted as the convergence result in Theorem 4 when K goes to $\infty$?
Without the adaptive step size mechanism designed in your paper, will the same convergence rate still hold under the same assumption? Will the same result hold with constant step size or for example $\gamma_t = 1/\sqrt{t}$?

---

> ### Author Response · Authors · 2020-11-18
> **Reply to AnonReviewer3**
>
> We thank the reviewer for their positive evaluation and their valuable input. We address below the concerns raised by the reviewer and anser their questions:
>
> 1.   Following the reviewer's recommendations, we completely restructured our paper and used the extra page available in the rebuttal to add an extra "Numerical Results" section. In this section, we added a series of numerical experiments for a Wasserstein GAN used to learn covariance matrices. This amounts to the (unbounded, non-Lipschitz) min-max objective
> $$
> \mathcal{L}(\theta,\phi) = \mathbb{E}_x[x^\top \theta x] - \mathbb{E}_z[z^\top \theta^\top \phi \theta z]
> $$
> with $z$ following a centered normal distribution with covariance matrix $I$, whereas the data samples x are drawn i.i.d. from a centered Gaussian with (unknown) covariance.
>
> For the experiments, we compared our proposed method (AdaProx) to the EG and BL algorithms, tracing the square norm of the problem's defining vector field to measure convergence; as in the case of bilinear games, AdaProx gave the faster convergence rates.
>
> 2. The elements of $A$ were drawn i.i.d. (component-wise) from a centered Gaussian with unit covariance; we now provide all the relevant details in the new "Numerical Experiments" section of our paper.
>
> 3. We respectfully disagree that our convergence rate analysis is similar in any way to that of the AMP algorithm of Antonakopoulos et al. (2019). Quite the contrary, the AMP algorithm has a "halving" mechanism which is used to mitigate the successive difference quotient that estimates the problem's Lipschitz (or Bregman) constant. Because this does not use any historical gradient information, the analysis of AMP relies on the standard mirror-prox template modulo a small adjustment for this halving mechanism. On the contrary, the proposed AdaProx method relies on the *aggregation* of past gradients to achieve rate interpolation between different problem classes; as such, it requires completely different tools and techniques relative to AMP (see appendices C and D).
>
> 4. Regarding the reviewer's last question: does the reviewer mean Theorem 3 instead of 4? If so, then, if we understand the question correctly, rate interpolation does not hold with a $1/\sqrt{t}$ step-size. We are also not sure what role $K$ play would play in this (since this is the strong convexity constant of the regularizer, it is not part of the problem's primitives).
>
> Thanks again for your time and positive evaluation – and we remain at your disposal for any clarification.

---

### Official Review · AnonReviewer5 · 2020-11-07
**The paper is interesting but its relevance to ICLR is not clear and I have strong doubts about the practicality of the proposed stepsize**

**Rating:** 5
**Confidence:** 4

**Review:**

### Summary.
This work proposed a stepsize for the extragradient/mirror-prox method that works both in smooth and non-smooth settings. The stepsize is based on the empirical values of gradient/operator differences, and mirror maps are used to allow for non-euclidian geometry such as KL divergence. The paper offers us three convergence results: 1 bound for the extragradient update (no mirror map) and 2 bounds for the mirror-prox update (convergence of iterates without a rate and convergence of the restricted gap with rates). The theory is followed by simple randomly-generated problems, which is ok for justifying the theory but can't serve a significant contribution on its own.
Overall, I enjoyed reading this paper and would recommend it for acceptance if not for two issues: 1) it seems to have low relevance to ICLR and I think it will not get much attention; 2) the stepsize is by definition upper-bounded by 1, and therefore will be suboptimal whenever the optimal stepsize is larger than 1.

### Relevance
The paper starts off with a number of interesting applications: GANs, reinforcement learning and adversarial learning. Unfortunately, it's never explained how the proposed algorithms are relevant to these applications. I think it's fine that the theory is derived for monotone operators, since many algorithms designed for convex objectives often work on nonconvex problems as well. However, unlike the algorithm of Bach and Levy, the one presented here does not allow for stochastic gradients, which is crucial in practice. And it seems to me that the motivation in the introduction is disconnected from the rest of the paper, for instance, it's not obvious why the projection step in (EG) plays any role.
I'm particularly confused by the importance of the "Non-Lips. / Unbounded" property of the stepsize. For instance, Table 1 mentions as an application the D-design, which is not a minmax problem, it's a minimization problem [1] (please correct me if you meant something else). Another example is the KL minimization problem, which is again not a minmax problem, and I do not see why it's of any relevance to the conference. I also think that this problem can be solved by methods from [1, 2] (depending on if it's smooth or not) Finally, the problem in Example 5.3 is also not motivated in this paper. Can the authors elaborate on its importance or give an example of some intereseting problem with minmax/VI structure?

### Suboptimality of the stepsize
The constant 1 in the stepsize, namely in the denominator in Equation (8), makes me skeptical about the method's universality. Due to this choice, we always have gamma_t <= 1, which might not be a good choice. Let's say we take an objective with smoothness constant L=10^(-4), then one can use stepsize gamma_t = 10^4, so the proposed stepsize will be quite suboptimal. I think the reason the authors used constant 1 is that it allows them to claim that the method is parameter-agnostic, but the cost of this choice is that it's not really adaptive. To make it adaptive to the problem properties (smoothness or gradient bound), one needs to have some problem-dependent constant in the denominator, but this means we exchanged one parameter (the stepsize itself) for another (the constant in the denominator).

The fact that the stepsize monotonically decreases is another flaw. RMSprop and Adam have been more successful than Adagrad exactly for this reason, and at least for GRAAL the stepsize can increase even close to the end of the training. In terms of GANs, it has been mentioned in several works, for example [3, 4, 5] that extragradient updates work better when combined with Adam stepsizes.

This is why I have strong doubts about the universality of the stepsize and tend to suggest rejection of this paper.

#### Minor issues:
What is a "divergent operator"? This term does not seem to be introduced anywhere. Moreover, one of the references for its importance (the authors mention [1, 21, 55]), [21], has a wrong title (the correct one is "An Adaptive Proximal Method for Variational Inequalities").

The experiments are only a small contribution as they are only given to support the theory, but I still think it would be better if the authors provided their code.

In Lemma 1 of the main part, the authors use the notation $\gamma_{\infty}$, but in the appendix, they use $\alpha$. Please make the notation consistent.

After formulating Lemma 1, the authors write "UniProx enjoys an accelerated O(1/T) rate of convergence under (MS)". I find the word "accelerated" to be confusing since O(1/T) is the standard rate under (MS), I suggest the authors clarify this place.

Typos:
When going from Equation (D.19) to Equation (D.21), you forgot to divide the right-hand side by $\sum_{t=1}^T \gamma_t$.
Page 18: "we readily get that under (MB) we get that:" -> "we readily get that under (MB):"
Page 22: "Therefore, y letting T" -> "by letting"
In the proof of Lemma F.1: "The lemma will proved by induction" -> "will be proved"

[1] Lu et al., "Relatively-Smooth Convex Optimization by First-Order Methods, and Applications"
[2] Lu, ""Relative-Continuity" for Non-Lipschitz Non-Smooth Convex Optimization using Stochastic (or Deterministic) Mirror Descent"
[3] Gidel et al., "A variational inequality perspective on generative adversarial networks"
[4] Mishchenko et al., "Revisiting Stochastic Extragradient"
[5] Chavdarova et al., "Reducing noise in GAN training with variance reduced extragradient"

---

> ### Author Response · Authors · 2020-11-18
> **Reply to major points**
>
> We would first like to thank the reviewer for their time and thorough review. While we disagree with some of the reviewer's conclusions, we sincerely appreciate their input and remarks, and we are happy they enjoyed our paper and found it interesting. We have uploaded a drastically revised version of our paper, with all relevant changes highlighted in blue, and we reply here to the reviewer's major concerns.
>
> **1. Relevance:** We respectfully disagree with the reviewer here and we note that this concern was not shared by any of the other expert reviewers. There is a considerable corpus of theoretical ICLR papers on min-max optimization and games, see e.g., Daskalakis, et al (2018), Gidel et al (2019), Mertikopoulos et al (2019), Yadav et al (2018), and the numerous submissions this year (a quick search on openreview reveals more than 15 papers). Given the flurry of activity on min-max optimization, we do not think that the relevance of theory papers on the topic can be questioned.
>
> If we understand correctly, the reviewer's main concern seems to be that our paper did not provide a sufficient range of examples for the unbounded/non-Lipschitz framework. The reason we did not elaborate was that many of the problems we mentioned (D-design, Poisson inverse problems, etc.), are usually subject to sparsity (or similar) constraints that are treated by passing to a Lagrangian of the form
> $$
>  \min_x \max_y L(x,y) = f(x) - y g(x)
> $$
>  where $f$ is the problem's primal objective (typically a KL loss term), $g$ represents the problem's constraints, and $y$ is a Lagrange multiplier. These problems are unbounded in $y$ and singular in $x$, so they are prime examples of our framework - and very common in statistical learning to boot.
>
> This is of course just one class of relevant examples. To expand our paper's scope, we zoomed in on a different class of VI/equilibrium problems from distributed computing that require the full capacity of the unbounded/non-Lipschitz setting. In our paper's revision, this example forms the core of Section 2.3, and we explore it further in Example 5.3 (also revised).
>
> Finally, regarding GANs, we added a series of numerical experiments on a Wasserstein GAN for learning covariance matrices, following the ICLR paper of Daskalakis et al. (2018).
>
> We are confident that the above provides more than sufficient evidence for the relevance of our paper to the community.
>
>
> **2. On the method's step-size:** We believe there may have been some confusion regarding the focus of our paper. To clarify, "adaptivity" can mean several things: adaptivity to *different* problem classes means achieving order-optimal rates for each class; adaptivity within a *given* problem class means achieving optimal "in-class" rates. Our focus was clearly the former, not the latter; as such, the reviewer's concern seems somewhat unwarranted (and it was not echoed by any of the other reviewers).
>
> We believe this misunderstanding may be due to our use of the term "universality" which, admittedly, is often used to characterize "intra-class" rather than "inter-class" adaptivity. Thus, to make things absolutely clear, we changed all occurrences of "universal" to "adaptive", "optimal" to "order-optimal", and our method's name from "UniProx" to "AdaProx". We trust that this removes any lingering doubts about the focus of our paper.
>
> Now, what the reviewer states about making the algorithm adaptive to "intra-class" parameters is correct: we could indeed trade in the "parameter-agnostic" property by introducing a suitable constant. However, since we care primarily about coarse-grained, worst-case instances, we sided with the "parameter-agnostic" part of the trade-off – but, of course, our analysis goes through with any positive constant instead of $1$ in the denominator of $\gamma_t$.
>
> Now, concerning methods with a non-decreasing step-size. First, we are not aware of *any* method achieving *inter-class* adaptation without a step mitigation mechanism. GRAAL indeed has a "not always decreasing" step-size, but it offers no inter-class guarantees, so the benefits of a non-decreasing step-size for inter-class adaptation are not clear. Also, GRAAL incorporates a moving average mechanism which lags behind the algorithm's iterates; as a result, it is not entirely clear how "aggressive" GRAAL steps actually are.
>
> Regarding Adam/RMSprop: first, we should clarify that these are completely different algorithms, not "just" step-size policies. Moreover, the evidence cited by the reviewer for Adam and RMSprop is purely empirical and only concerns GANs: the cited papers do not provide any proof of convergence for Adam or RMSprop for general min-max games, let alone any interpolation results grounded in theory. This is a very hard problem on which very little progress has been made over the last years, so it is not reasonable to expect our paper to answer this too.
>
> We hope and trust that the above resolves the reviewer's main concerns.

---

> > ### Comment · AnonReviewer5 · 2020-11-24
> > **The issues remain that: 1) the addressed aspects of minmax problems seem to have low relevance, and 2) the stepsize can be extremely suboptimal**
> >
> > Thank you for your response. I am a little bit worried that for both points you wrote that the other reviewers did not raise the same issues. Firstly, it is my responsibility as a reviewer to point out issues even if no one else mentions. If there was a mistake, it would not matter how many reviewers mentioned it, right? The same principle should apply in other cases too. Secondly, for both issues, I find that Reviewer 4 actually has similar points (almost identical for the first one). I hope that by pointing out where I agree with Reviewer 4, I can clarify what my concerns are exactly.
> > 1. The authors incorrectly write that "this concern was not shared by any of the other expert reviewers". I think that the first comment of Reviewer 4 is about exactly the same issue: "No concrete important problems intractable before by the standard extragradient but tractable by the proposed version".
> > I strongly agree that minmax problem itself can be very interesting and the mentioned papers/submissions support this. I do not agree that any arbitrary aspect of minmax problem is interesting. The aspects of this paper (singularity, mirror map, etc.) do not seem interesting.
> > I appreciate the effort to add the new experiment for covariance matrix learning, but note that this problem does not require Bregman divergences, does not have singularities and overall does not look like a good fit for your method.
> > I hope my comment makes it clear why exactly I have serious concerns about applications.
> >
> > 2. Reviewer 4 seems to agree with me on that issue: "Quite bad constants." and "I feel like the authors unconsciously push me to suspect that the proposed algorithm will not perform well in most cases." The reasons are different but the conclusion is the same: the method will perform well.
> > To avoid further confusion, let me simplify my point about stepsizes to one sentence: the method is going to be slow. It is slow because the stepsize is by definition upper bounded by 1, which can be arbitrarily suboptimal, and because the stepsize is decreasing, unlike the one in GRAAL. You make a point that this is not your focus, but I do not agree that this can be ever ignored: since it's not the only method for minmax problems, its efficiency is one of the main things that we need to pay attention to. Of course, it is not the only aspect and the interpolation property might be nice in some situations, but when it comes to practice, method's efficiency is the main property. Since GRAAL can actually adapt to the problem's properties and can increase the stepsize, I expect it to be faster despite using a moving average mechanism, not to mention that the rates you prove are for the averaged iterate in Equation (4). To conclude, I do not acknowledge that your method has a property that GRAAL doesn't and I consider this as a strength, but it is, in my opinion, a minor strength while the weakness is significant.
> >
> > I agree with your points about Adam and RMSprop. I do not expect you to produce a theory for them taking into account there is little understanding of why they work for minimization. All I want to say is that your method is unlikely to have an influence on neural network training due to the efficacy of Adam. I do not suggest trying to get a theory for it but rather point it potential issues with applying your method in practice.

---

> > > ### Author Response · Authors · 2020-11-24
> > > **Thanks for your continued input**
> > >
> > > We do our best to respond to your points below, but please keep in mind that your reply came only a few hours before the end of the discussion period, so we cannot hope to be as thorough as we would have otherwise liked.
> > >
> > > 1. Regarding other comments: we only wished to clarify the source of the concern, to make sure that we reply in context.
> > >
> > > 2. Regarding concrete problems that "were intractable before by the standard extragradient". Please note that we devoted an entire section (Section 2.3) to an application from high-performance computing clusters with FCFS processing: this problem is both singular and non-smooth, so it cannot be treated by existing extra-gradient methods. We also provided in our response above a class of examples that arise in the Lagrangian formulation of D-design and/or Poisson inverse problems under constraints. Your response did not acknowledge any of the above, even though none of the standard extragradient methods (Bach-Levy, GRAAL, etc.) can tackle these concrete problems.
> > >
> > > 3. Regarding experiments: we focused on covariance learning because this is a standard benchmark, but we would be happy to switch focus if you wish. [Though, of course, we cannot provide new numerical results in the few hours that remain until the discussion period is closed to authors]
> > >
> > > 4. On whether the provided constants are "bad" or not: as we explain in our response to AnonReviewer4, the constants that we provide actually represent an improvement over the corresponding Bach-Levy constants (which become infinite if the problem does not have a bounded domain or bounded gradients).
> > >
> > > 5. We do not understand the phrase "I do not acknowledge that your method has a property that GRAAL doesn't". Is this a typo? If this is not the case, we would kindly ask the reviewer to explain how GRAAL can provide rate interpolation between smooth and non-smooth problems, or how it can deal with problems with singularities (like the computing cluster example of Section 2.3).
> > >
> > > 6. Regarding the comparison to GRAAL: in the paper of Malitsky (2020), the only theoretical guarantee for GRAAL (or, rather, aGRAAL since GRAAL is not adaptive) is given by Eq. (39) in the arxiv version of the paper. This rate is of the form $O(M/\sum_{k=1}^{n} \lambda_k)$, where $M$ is a constant that depends on the algorithm's initialization and $\lambda_{k}$ is the method's step-size. In turn, $\lambda_k$ is bounded from below by $c/L^2$ where $c$ is a constant that depends on user-tuned parameters and $L$ is the problem's Lipschitz constant in a set containing the initialization of the algorithm and a solution point. As a result, in problems with *very large* Lipschitz constants (say $L=10^4$), GRAAL will also have a "bad" $O(L^2)$ constant relative to standard extra-gradient which has an $O(L)$ dependence. In this regard, there is no theoretical ground to suggest that GRAAL performs any better than AdaProx in problems with very large Lipschitz constants: these problems are the primary focus of our paper, so the reviewer's criticism that "[our] method is going to be slow" without further qualifications is unfounded. Specifically, we are willing to acknowledge that aGraal may perform better than AdaProx in problems with a *very small* Lipschitz constant; however, we expect that the reviewer will also acknowledge in turn that aGraal may perform worse than AdaProx in problems with a *very large* Lipschitz constant. We feel that this observation alone is of sufficient interest - even before discussing the rate interpolation and "beyond Lipschitz" guarantees of AdaProx.
> > >
> > > 7. Regarding the upper bound of our method's step-size: related to the above, the reviewer is also claiming as a weakness the fact that the step-size of the method is upper bounded by $1$. As we replied elsewhere (specifically to AnonReviewer1), this can be mitigated efficiently by restarting the algorithm and using a doubling trick to control the length of the restart window. This would effectively give the adaptation property that AnonReviewer5 is seeking: possibly increasing step-sizes near the problem's solution if the problem's smoothness permits it. Alternatively, one can use a line-search mechanism as in the works by Gasnikov and co-authors: these are all modules that can be fitted to an underlying AdaProx chassis. Finally, it is also possible to use an arbitrary constant instead of $1$: this could considerably speed up the algorithm at the cost of introducing a tunable parameter (though note that aGRAAL *also* includes the tunable parameter $\bar\lambda$).
> > >
> > > All these are very promising directions for future research, but it is not possible to fit everything in a single 8-page paper. Indeed, we feel that these issues showcase a range of open questions that can be tackled by researchers following up on our analysis.
> > >
> > > Again, we would like to thank the reviewer for their continued input, and we hope that the above replies and clarifications alleviate their remaining concerns.

---

> > > > ### Comment · AnonReviewer5 · 2020-11-24
> > > > **Further clarifications**
> > > >
> > > > Thank you for the prompt response, sorry that you didn't have much time to write it.
> > > > 1. Great.
> > > > 2. Thank you for providing the additional problem. This looks like a very classical convex optimization problem, in fact, one of the references, [5], is from 1992. I find that this problem, just as many others for which the algorithm is suitable, are of higher relevance to an optimization journal (like SIAM) than to a conference on learning.
> > > > I acknowledge that you can also address D-design with constraints. While it's not clear to me that this problem is either smooth or has bounded gradients (if y is unbounded, it does not seem to be ), the more important issue is that I don't see why this problem is relevant.
> > > > 3. I think I made it clear in my original review what problems I find highly relevant ("The paper starts off with a number of interesting applications: GANs, reinforcement learning and adversarial learning"). I'm sorry that the authors didn't have time to run additional experiments because of me, but my role is only to point out the merits and flaws of the papers.
> > > > 4. Thank you for repeating this argument to me. I understand that the method of Bach and Levy has even worse constants, but this does not explain why your method will be fast.
> > > > 5. This was a typo, I wanted to write that "I acknowledge that your method has a property that GRAAL doesn't".
> > > > 6. Indeed, we should refer to aGRAAL rather than GRAAL. I only made a point about the practical behavior of aGRAAL, I did not intend to compare the complexities.
> > > > 7. I do not see why the proposed approach would preserve the rate interpolation property. It's easy to say that you can make an extension but another thing is to actually do it. Anyhow, since the authors say it's only a future direction,  this should not affect the discussion of the paper in its current form.
> > > >
> > > >
> > > > Unless I missed it, it seems that the authors ignored my point in the initial review about stochastic gradients. There is a mention of this in the response to Reviewer 1, but it doesn't address the lack of theory for this case.

---

> > > > > ### Author Response · Authors · 2020-11-25
> > > > > **Clarifications**
> > > > >
> > > > > Thanks again for the prompt input, we are in turn providing replies below.
> > > > > - For your points (2) and (3), taken together: "[It's] not clear to me that this problem is either smooth or has bounded gradients (if $y$ is unbounded, it does not seem to be)". Indeed, the lack of bounded gradients and smoothness in the classical sense is precisely the problem: this is exactly where the Finsler regularity formalism and guarantees kick in, it's Theorem 2 that provides the answer for this case, not Theorem 1. We explain this in detail for the HPC problem in Section 5; since D-design and Poisson inverse problems have logarithmic singularities in the objective, an analogous setup works for these problems as well. As for the relevance of D-design and Poisson inverse problems, they are both of key importance in statistical learning and imaging respectively, see for example the 2020 ICLR spotlight of Antonakopoulos et al. We would of course be happy to provide a detailed presentation of these problems in addition to the HPC problem, and we commit to do so in a subsequent revision, but there is no time to update our manuscript at this stage.
> > > > > - For your points (4) and (6), taken together: we used Bach-Levy as a benchmark because it is the state-of-the-art method for achieving rate interpolation between different regimes - i.e., interpolate between $O(1/T)$ and $O(1/\sqrt{T})$. Methods that are designed to work for a wider class of problems (like Bach-Levy) might be less efficient in a subclass thereof when compared to methods that have been explicitly designed to solve problems from the narrower subclass (like GRAAL in this case). [By contrast, GRAAL has no rate guarantees in non-smooth problems when the non-smooth component cannot be offloaded to a backward proximal step that is easy to compute and does not offer any rate interpolation guarantees.] This is also why we said in our reply that GRAAL can outperform AdaProx in problems with a very small Lipschitz constant - but, in turn, AdaProx can outperform GRAAL in problems with a very large Lipschitz constant.
> > > > > - For your point (5): great.
> > > > > - For stochastic gradients: this is also an aspect in which the AdaProx approach can help. Roughly speaking, in the presence of noise, each update introduces a $\Theta(1)$ term in the sum that appears in the denominator of the proposed step-size policy. As a result, the AdaProx step-size will behave asymptotically as $\gamma_t = \Theta(1/\sqrt{t})$, which is the optimal step-size for stochastic problems, and leads to an $O(1/\sqrt{T})$ convergence rate in expectation. This is consistent with the numerical experiments presented in Section 7 but a full-scale theoretical treatment is beyond the scope of our paper.
> > > > > - For your point (7): indeed, this is why we stated this approach as a future direction; we feel there is appreciable value in providing concrete directions for follow-up research in such a vigorous field.
> > > > >
> > > > > Thanks again for your time and remarks!

---

> ### Author Response · Authors · 2020-11-18
> **Reply to minor points**
>
> On the reviewer's more minor comments:
> - An operator is "divergent" if it develops a singularity at a boundary point of its domain. We removed this terminology and changed it to "singular/singularities".
> - $\gamma_\infty$ has been fixed.
> - Code: sure, will do.
> - Use of "accelerated": removed throughout.
> - Typos: fixed.
>
> Thanks again for your time and input – needless to say, we are at your disposal for any further clarification.

---

> > ### Comment · AnonReviewer5 · 2020-11-24
> > **Acknowledged**
> >
> > Thanks for fixing this.

---

### Decision · Program_Chairs · 2021-01-07
**Final Decision**

**Decision:**

Accept (Poster)

**Comment:**

The paper introduces a new step size rule for the extragradient/mirror-prox algorithm, building upon and improving the results of Bach & Levy for the deterministic convex-concave setups. The proposed adaptation of EG/Mirror-prox -- dubbed AdaProx in the submitted paper -- has the rate interpolation property, which means that it provides order-optimal rates for both smooth and nonsmooth problems, without any knowledge of the problem class or the problem parameters for the input instance. The paper also demonstrates that the same algorithm can handle certain barrier-based problems, using regularizers based on the Finsler metric.

The consensus of the reviews was that the theory presented in the paper is solid and interesting. The main concerns shared by a subset of the reviews were regarding the practical usefulness of the proposed method. In particular, the method exhibits large constants in the convergence bounds and cannot handle stochastic setups. Further, the empirical evidence provided in the paper was deemed insufficient to demonstrate the algorithm's competitiveness on learning problems. If possible, the authors are advised to provide more convincing empirical results in a revised version, or, alternatively, to tone down the claims regarding the practical performance of the method.